# Genome-Wide Identification of the *CER* Gene Family and Significant Features in Climate Adaptation of *Castanea mollissima*

**DOI:** 10.3390/ijms232416202

**Published:** 2022-12-19

**Authors:** Shuqing Zhao, Xinghua Nie, Xueqing Liu, Biyao Wang, Song Liu, Ling Qin, Yu Xing

**Affiliations:** College of Plant Science and Technology, Beijing Key Laboratory for Agricultural Application and New Technique, Beijing University of Agriculture, Beijing 102206, China

**Keywords:** *Castanea mollissima*, *CER* gene family, drought stress, epicuticular wax layer thickness, expression

## Abstract

The plant cuticle is the outermost layer of the aerial organs and an important barrier against biotic and abiotic stresses. The climate varies greatly between the north and south of China, with large differences in temperature and humidity, but Chinese chestnut is found in both regions. This study investigated the relationship between the wax layer of chestnut leaves and environmental adaptation. Firstly, semi-thin sections were used to verify that there is a significant difference in the thickness of the epicuticular wax layer between wild chestnut leaves in northwest and southeast China. Secondly, a whole-genome selective sweep was used to resequence wild chestnut samples from two typical regional populations, and significant genetic divergence was identified between the two populations in the *CmCER1-1*, *CmCER1-5* and *CmCER3* genes. Thirty-four *CER* genes were identified in the whole chestnut genome, and a series of predictive analyses were performed on the identified *CmCER* genes. The expression patterns of *CmCER* genes were classified into three trends—upregulation, upregulation followed by downregulation and continuous downregulation—when chestnut seedlings were treated with drought stress. Analysis of cultivars from two resource beds in Beijing and Liyang showed that the wax layer of the northern variety was thicker than that of the southern variety. For the Y-2 (*Castanea mollissima* genome sequencing material) cultivar, there were significant differences in the expression of *CmCER1-1*, *CmCER1-5* and *CmCER3* between the southern variety and the northern one-year-grafted variety. Therefore, this study suggests that the *CER* family genes play a role in environmental adaptations in chestnut, laying the foundation for further exploration of *CmCER* genes. It also demonstrates the importance of studying the adaptation of Chinese chestnut wax biosynthesis to the southern and northern environments.

## 1. Introduction

Drought has become a global problem, with models predicting that drought conditions will increase over the next 90 years due to decreasing precipitation and evaporation of water [1,2]. China’s complex topography, diverse climate types and large zonal differences also make drought disasters more severe. It is generally believed that the Qinling–Huaihe line is the dividing line between the subtropical monsoon climate/region (south) and the warm temperate monsoon climate/region (north). The average annual precipitation is 400–800 mm in the north, and the annual average precipitation in the south is above 800 mm, with a drier and colder climate in the northern area than in the southern area [3]. The Chinese chestnut (*Castanea mollissima* Bl.) is widely distributed in China and exists in various ecological types [4] from Jilin province in the north to Hainan province in the south. Compared with *Castanea seguinii* and *Castanea henryi*, the Chinese chestnut is cultivated in the north of the Qinling mountains in China for its excellent fruit quality, adaptability and comprehensive edible value.

The plant cuticle is a composite structure consisting of a keratinous macromolecular scaffold and a variety of organic solvent soluble lipids, collectively known as waxes [5]. The biosynthesis of waxes consists of three stages: the first stage is the ab initio synthesis of C16 and C18 fatty acids from plastids, which act as central intermediates for all lipids. The second stage is the extension of C16 and C18 fatty acids into very-long-chain fatty acids (VLCFAs) with C20–C34 in the endoplasmic reticulum (ER). Following extension, the VLCFAs are modified to create major waxy components, including esters, aldehydes, alkanes, ketones and alcohols, via the decarbonylation pathway and the acyl reduction pathway [6]. It has been demonstrated in previous studies that wax synthesis in plants is controlled by the *CER* (eceriferum) family of genes.

Members of the *CER* family genes have been extensively studied. There are 17 known *CER* family genes in the Arabidopsis thaliana Information Resource (TAIR) database. *CER* family genes have been identified in the genomes of sunflower (*Helianthus annuus*) [7], *Ziziphus jujube* [8], apple (*Malus domestica*) [9] and passion fruit (*Passiflora edulis*) [10]. The *CER1* gene encodes decarbonylase, an enzyme that converts long-chain aldehydes into alkanes, a key step in wax biosynthesis [11]. The complex of *CER3* and *CER1* together catalyze the formation of alkanes from VLCFA-CoA [12]. *CER3* expression is associated with *CER16*, and inactivation of *CER16* inhibits *CER3* expression [13]. The *CER2* protein is a regulatory protein of the fatty acid elongase complex. *CER2* accumulates in a variety of organs and cell types. *CER2* expression is epidermis-specific and *CER2* acts on C28 elongation [14,15,16]. *CER2* and *CER6* act together on C30 synthesis [17]. Both *CER26* and *CER2* are involved in the formation of extra-long-chain fatty acids and act as mechanisms of fatty acid elongation in wax synthesis [18]. *CER4* encodes the alcohol-forming fatty acyl coenzyme A reductase (FAR) and is expressed in leaves, stems, flowers and roots [19]. *CER4* and *CER17* share a role in the synthesis of ultra-long chain monounsaturated alcohols [20]. *CER5* plays a role in surface wax secretion by working with WBC11 [21]. *CER6* is a key enzyme in the biosynthesis of straight lipids in the pollen shell and in the stem cuticle [22]. *CER60* is highly similar to *CER6* nucleotides and amino acids and is involved in saturated ultra-long-chain fatty acid biosynthesis [22,23]. Cuticular wax loading is regulated by enzymes encoded by *CER7* and *CER9*, with *CER7* regulating the exosomal core subunit of wax production during inflorescence development and the *CER9* gene encoding an E3 ubiquitin ligase that is a negative regulator of wax and keratin monomers [24,25]. *CER8* encodes acyl coenzyme A synthase, an enzyme that preferentially modifies the VLCFA of wax synthesis and keratin synthesis of long-chain (C(16)) fatty acids [26]. *CER10* plays an important role in the biosynthesis of C26–C30 fatty acids, primary alcohols and aldehydes [27]. *CER11* encodes a protein that catalyzes the dephosphorylation step in the secretion of extracellular matrix components in epidermal cells [28]. *CER13* is involved in the enzyme-specific message of wax ester formation [29].

When plants changed from an aquatic to terrestrial environment 450 million years ago, terrestrial plants were exposed to a variety of new environmental challenges, including desiccation, temperature changes, UV radiation exposure and mechanical damage [30,31]. To adapt to terrestrial life, terrestrial plants form cuticles on the surface of their aerial organs, unique structures that are the result of the ability of plants to evolve morphological and physiological traits to maintain water [5,30]. Drought is a major abiotic stress that affects plant growth and productivity and causes yield losses [32,33]. Studies have shown that epidermal waxes act as a protective barrier against environmental stresses and that various environmental factors also influence the biosynthesis, composition and morphology of waxes [34]. The accumulation of waxes is associated with the prevention of water loss and contributes to the drought tolerance of plants [35,36]. Under drought conditions, the wax content of *A. thaliana* leaves and stems increases [37]. Additionally, biochemically, changes in the ultra-long-chain fatty acid derivatives of *A. thaliana* have also been observed [38]. Drought significantly increases the alkane content in wheat lines [39]. Expression of *SlCER1* and *SlCER3* was significantly upregulated in one-month-old tomato plants under drought stress, and overexpression of *SlCER1-1* significantly increased very-long-chain (VLC) alkane biosynthesis and wax accumulation [40]. Total wax esters were significantly elevated in grapes under water deficit conditions and *CER1*, *CER3*, *CER2*, *CER4*, *CER10* and *WSD1* were upregulated in the grape wax biosynthesis pathway [41]. *MYB96* induced the transcription of genes involved in VLCFA biosynthesis such as *KCS1*, *KCS2* and *KCS6*, and was also secondary to *CER1* and *WBC11* transcription [36]. In a study on dinoflagellates, it was shown that alkanes are more sensitive to drought stress than other waxes and that the *BdCER1* gene plays different roles in different emergency pathways [42]. In transgenic cucumber, *CsCER1* influences the biosynthesis and drought resistance of VLC alkanes [42]. In apple, *MdCER1*, *MdCER2*, *MdCER3*, *MdCER4* and *MdCER8* are able to respond to drought in response to PEG induction [9].

A significant relationship between drought and *CER* genes has been confirmed in species such as tomato and grape, where *CER* genes influence wax biosynthesis, and in species such as apple, passion fruit and sunflower, for which the *CER* family was analyzed; however, the relationship between chestnut *CER* genes and drought stress has been less studied, and there is no systematic analysis of *CER* genes in Chinese chestnut. In the investigation of wild chestnut samples, we found that the thicknesses of the cuticles of the leaves in the north and south were different. At the same time, genes related to cuticle synthesis were found through genome-wide selective sweep analysis between the northern and southern populations of wild chestnut. Therefore, in this study, 34 *CER* genes from the *C. mollissima* genome were systematically characterized and analyzed through comparative bioinformatics. It was found that *CER* gene expression was induced by drought stress. This study provides a basis for further research on wax biosynthesis in chestnut leaves. We explored the importance of the *C. mollissima CER* gene family in adaptation to northern and southern climates.

## 2. Results

### 2.1. Analysis of the Thickness of the Cuticle in Wild Chinese Chestnut Samples Collected from Northern and Southern China

A total of 146 wild Chinese chestnut samples collected in Gansu, Shaanxi, Sichuan, Hubei, Anhui, Zhejiang, Jiangxi and Hunan were used to produce semi-thin sections, which were stained using oil red and then examined under a microscope. The staining results of all samples were measured using ImageJ and Graphpad Prism software. The wild samples were divided into two categories—northern and southern—according to the eight provinces where they were collected, to obtain box line plots of the epicuticular wax layer thickness of the wild species. For the wild Chinese chestnut samples, the epicuticular wax layer thickness of the northern samples was significantly (*p* < 0.0001) thicker than that of the southern samples (Figure 1a). The epicuticular wax layer thickness of the sample collected from Gansu was 2.56 times that of the sample collected from the Tianmu Mountains, Zhejiang (Figure 1b,c).

### 2.2. Genome-Wide Selective Sweep Analysis between the Populations of Wild Chinese Chestnut Collected from Northern and Southern China

To further explore the difference between the two populations, samples of wild Chinese chestnut from two typical regions (10 from Gansu and 10 from Tianmu Mountains) were collected and scanned by genome-wide selection sweeps (Figure 2). We identified three *CER* genes (*CmCER1-1*, *CmCER1-5* and *CmCER3*) with significant genetic differentiation between the two populations, suggesting their potential contribution to the adaptation of Chinese chestnut to different climates. Therefore, we speculate that the *CmCER* gene family plays an important role in adaptation to northern and southern climates.

### 2.3. Identification of CmCER Family Genes in Chinese Chestnut

We searched for 17 *AtCER* genes in *A. thaliana* and identified 34 *CmCER* gene family members in the Chinese chestnut genome. Furthermore, the physicochemical properties of 34 *CmCER* gene proteins were analyzed.

According to the analysis, the number of amino acids in *CmCER* genes ranged from 142 (*CmCER1-9*) to 1853 (*CmCER13-1*). The predicted molecular weights ranged from 16.50 (*CmCER1-9*) to 206.05 (*CmCER13*) kD, and the isoelectric points ranged from 5.28 (*CmCER2-1*) to 9.71 (*CmCER60-5*) (Table 1).

Subcellular localization showed that *CmCER1-1*, *CmCER1-2*, *CmCER1-3*, *CmCER1-4*, *CmCER1-6*, *CmCER1-7*, *CmCER1-8* and *CmCER1-9* were localized in vacuoles. *CmCER1-5* was localized to cell membranes and vacuoles. *CmCER2*, *CmCER6-1*, *CmCER6-2*, *CmCER60-2*, *CmCER60-3*, *CmCER60-4*, *CmCER60-5*, *CmCER60-7* and *CmCER60-8* were localized to the cytoplasm. *CmCER60-1* and *CmCER60-6* were localized to chloroplasts and cytoplasm. *CmCER4* was localized to the Golgi apparatus, and *CmCER5-1*, *CmCER5-2* and *CmCER9-2* were localized to the cell membrane. *CmCER7* was localized in chloroplasts and mitochondria. *CmCER8* was localized to peroxisomes. *CmCER9-1* was localized to cell membranes and nuclei. *CmCER10-1*, *CmCER11-1* and *CmCER13* were localized to cell nuclei. *CmCER17* was localized to mitochondria and the endoplasmic reticulum. *CmCER9-1* was localized to cell membranes and nuclei. CmCER10-2 was localized to cell membranes and the Golgi apparatus. *CmCER11-2* was localized to mitochondria and cell nuclei (Table 1).

### 2.4. Phylogenetic Analysis of CmCER Proteins in Chinese Chestnut

In order to evaluate the functional characteristics and evolutionary relationships of the *CmCER* gene family, this study used 17 *A. thaliana* and 34 *C. mollissima CmCER* genes encoding protein sequences for multiple comparisons, and constructed a phylogenetic tree using the MEGA11 software. Five genetically related groups were identified and the *CmCER* genes were named. The relationships between genes in the same cluster showing homologous CER protein sequences are shown. There is one *CmCER* gene (*CmCER2*) in CLAD1. There are six genes in CLAD2, including four *CmCER* genes, namely, *CmCER5-1*, *CmCER5-2*, *CmCER9-1* and *CmCER9-2*. There are 14 genes in CLAD3, including 11 *CmCER* genes, namely, *CmCER6-1*, *CmCER6-2*, *CmCER8*, *CmCER60-1*, *CmCER60-2*, *CmCER60-3*, *CmCER60-4*, *CmCER60-5*, *CmCER60-6*, *CmCER60-7* and *CmCER60-8*. There are 10 genes in CLAD4, including six *CmCER* genes: *CmCER4*, *CmCER7*, *CmCER10-1*, *CmCER10-2*, *CmCER11-1* and *CmCER11-2*. There are 18 genes in CLAD5, including 12 *CmCER* genes: *CmCER1-1*, *CmCER1-2*, *CmCER1-3*, *CmCER1-4*, *CmCER1-5*, *CmCER1-6*, *CmCER1-7*, *CmCER1-8*, *CmCER1-9*, *CmCER3*, *CmCER13* and *CmCER17* (Figure 3). *CmCER1* and *CmCER60* showed more gene expansion (Figure 3).

### 2.5. Analysis of Gene Motifs of CmCER Gene Family in Chinese Chestnut

Ten protein motifs were predicted in the *CmCER* genome based on *CmCER* gene phylogenetic relationships. *CmCER1-1*, *CmCER1-2*, *CmCER1-3*, *CmCER1-4*, *CmCER1-5*, *CmCER1-6*, *CmCER1-7*, *CmCER1-8*, *CmCER1-9* and *CmCER3* share six motifs. *CmCER6-1*, *CmCER6-2*, *CmCER60-1*, *CmCER60-2*, *CmCER60-3*, *CmCER60-4*, *CmCER60-5*, *CmCER60-6*, *CmCER60-7* and *CmCER60-8* share four motifs (Figure 4). The results indicate that there is a conserved motif similarity between each group, and that these specific conserved motifs are associated with the *CERs*’ functions.

### 2.6. Chromosomal Localization of CmCER Gene in Chinese Chestnut

Analysis of the location of *CmCER* genes on Chinese Chestnut chromosomes showed that the 34 *CmCER* genes were unevenly distributed among the 12 chromosomes of Chinese Chestnut. Specifically, chromosomes 3, 7 and 10 contained five *CmCER* genes, chromosomes 1, 8, 9 and 11 contained three *CmCER* genes, chromosomes 2, 5 and 12 contained two *CmCER* genes, chromosome 6 contained only one *CmCER* gene and chromosome 4 contained no *CmCER* genes (Figure 5).

### 2.7. Chromosomal Localization of CmCER Gene in Chinese Chestnut

The base sequence of the first 2000 bp of the Chinese Chestnut start codon was selected to predict cis-acting elements. The results showed that 34 *CmCER* genes contained multiple cis-acting elements. Three cis-elements were analyzed. The first was hormone-response elements, including abscisic acid response, auxin response, gibberellin response, MeJA response and salicylic acid response. Among the 34 members, 7 were induced by one hormone and 7 by four hormones, 8 were induced by two hormones and 12 were induced by three hormones. Secondly, stress-response elements, including light induction, low-temperature induction and stress defense mechanism were analyzed. Thirty-four *CmCER* genes contain light-response elements, 12 *CmCER* genes respond to the stress defense mechanism, and 12 *CmCER* genes respond to low temperature. Finally, MYB binding was analyzed, including the flavonoid biosynthesis binding site, drought inducibility and the MYBHv1 binding site. Only 22 genes have MYB binding sites (Figure 6 and Appendix A). These cis-acting elements may directly determine the ability of *CmCER* gene family members to respond to stress conditions.

### 2.8. Changes of CmCER Gene Expression Level in Chinese Chestnut Seedlings under Drought Treatment

To investigate how the *CmCER* family genes respond to drought changes, Chinese chestnut seedlings were treated under drought conditions for different lengths of time. Expression levels of the *CmCER* family genes were determined by qPT-PCR. Thirty-two genes were activated under drought treatment. The expression levels of most *CmCER* genes increased with increasing drought treatment time. The expression level of three *CmCER* genes showed a trend of rising first and then decreasing. Eight *CmCER* genes showed a continuous downward trend in their expression levels (Figure 7a). The results showed that the expression levels of most *CER* genes were upregulated in response to drought stress.

In relation to evolutionary kinship analysis, five genes in the three expression patterns showing a rising and falling trend are in CLAD 2 and CLAD 5 of the evolutionary tree. The seven genes in the continuously decreasing expression pattern are in CLAD 1, CLAD 3, CLAD 4 and CLAD 5. The other genes are in CLAD 2, CLAD 3, CLAD 4 and CLAD 5 of the evolutionary tree.

In order to test whether the climate in the south and north has an impact on the epicuticular wax layer thickness, we selected varieties to compare the cuticle thickness from two germplasm repositories in the south (Longtan forestry nursery, Liyang, Jiangsu 119°27′10.106″ E 31°19′55.320″ N) and north (Chestnut Experiment Station, Huairou, Beijing 116°28′26.821″ E 40°25′14.614″ N). The varieties from the two germplasm repositories in Beijing and Liyang were divided into two major categories according to their places of origin, and box line diagrams of cuticle thicknesses of the northern and southern varieties were obtained. The samples from both the Beijing and Liyang germplasm repositories showed that the epicuticular wax layer thickness of the northern varieties was significantly (*p* < 0.01) thicker than that of the southern types (Figure 7b). Climatic differences between the north and the south result in different degrees of drought faced by Chinese chestnut. To compare the phenotypes of epicuticular wax layer thickness of the same variety under different climates (north and south), we used the same rootstock selected from wild Chinese chestnut Y-2 (*Castanea mollissima* genome sequencing material) for simultaneous grafting in both Hubei (where the variety originated) and Beijing, and took annual grafting material from both locations. The results showed that the thickness of the epicuticular wax layer of the grafted Y-2 leaves in Beijing (north) was significantly (*p* < 0.05) thicker than that of the Y-2 leaves in Hubei (south) (Figure 7c). The expression of *CmCER1*-1, *CmCER1*-5 and *CmCER3* was significantly different, and the expression of the grafted Y-2 in the north was significantly higher than that in the south (Figure 7d).

## 3. Discussion

The cuticle is an important barrier that covers the aerial epidermis of terrestrial plants and protects them from external environmental stresses [5]. In addition to enabling plants to survive in a dry environment, the cuticle also resists other biotic and abiotic stresses [43]. In a study of yellowhorn (*Xanthoceras sorbifolium* Bunge), a comprehensive analysis of leaf anatomy and cuticle properties was conducted on germplasm samples, revealing marked differences in leaf wax content and indicating that epidermal wax content is related to drought resistance. Thus, wax content is used as an important indicator for the screening of drought-resistant germplasm [44]. The differences in temperature and humidity between the south and north of China and the wide distribution of chestnut in China are also related to the drought tolerance of chestnut. By post-staining analysis of sections of Chinese chestnut samples from the north and south, it was determined that there were differences in cuticle thickness between the two regions due to their differing degrees of drought. A genome-wide selective sweep of wild Chinese chestnut samples from two population classes—NW and SE—led to the hypothesis that the *CmCERs* family of genes plays an important role in adaptation to northern and southern climates.

Drought tolerance is the ability of a plant to tolerate low water potential and maintain a certain level of physiological activity, growth and development. It is a complex quantitative trait in which the plant interacts with the drought environment. Drought tolerance in plants is influenced by several gene families; the ERECTA proteins belong to the protein kinase superfamily and the serine/threonine protein kinase family, which influence plant drought tolerance by affecting stomatal development, pathogen defense and phytohormone perception [45]. The *DREB* genes, which belong to the *AP2/EREBP* family, are involved in the control of non-ABA-dependent drought stress responses in combination with plant-specific, stress-regulated transcription factors [46]. The *Asr* gene family plays an important role in plant adaptation to drought, especially the *Asr1* gene, which is expressed in a housekeeping manner, and small changes in the protein can have a serious impact on plant adaptation [47]. Although other gene families have also been associated with drought tolerance, in this study the focus on the *CER* gene family was due to the significant divergence of *CERs* genes between the NW and SE populations identified in the whole-genome resequencing sweep analysis.

Genome-wide identification of *CER* family genes has been reported in different plants, including the identification of 37 *CER* genes and their homologs in the sunflower genome [7], 29 *ZjCER* genes in the date palm genome [8], 10 *AtCER* genes in the apple *MdCER* family analysis [9], and 10 *MdCER* genes in the passion fruit; in the *PeCER* gene family, 34 *PeCER* genes have been identified [10]. In this study, 34 *CmCER* genes were identified in the Chinese chestnut genome. *CER* family genes have shown varying degrees of gene expansion in sunflower, sour date, passion fruit and chestnut (the species of this study), influenced by different genetic replication events to evolve in a species-specific manner. We constructed a phylogenetic tree of the *AtCER* gene family and the *CmCER* gene family and identified five genetically related groups, with *CmCER1* showing more gene expansions. This may be due to tandem and dispersed duplication, with segmentation and tandem duplication playing an important role in the generation and maintenance of gene families. Gene duplication underlies the functional divergence that occurs in homologous genes and is influenced by different genetic duplication events to evolve in a species-specific manner, with *CER* homologs in different species receiving strong purifying selection and being functionally conserved. For example, in the crustacean Mongolian oak (*Quercus mongolica*), the evolutionary relationship between the *CER1* gene and drought stress response was analyzed. The homologous gene copy numbers of different genes identified in the epidermal wax biosynthetic pathway were species-specific, and it was found that the *CER1* homologous gene was significantly amplified in the wax biosynthetic pathway. The Mongolian oak has a wide distribution range and the amplification of the *CER1* homologous gene contributes to an improved response to drought stress [48]. The replication pattern of *CER* genes in Chinese chestnut needs further analysis.

The *CER* family genes of sunflower, date palm, apple and passion fruit are all located on different chromosomes, which may be due to their involvement in various functions, resulting in genes from the same family being distributed on different chromosomes. Conservative motif analysis of differences in protein function predicted 10 protein motifs. The *CmCER* gene family is highly conserved throughout evolution and plays an important role in wax biosynthesis. Our results are not consistent with Rizwan et al. Predictions of cis-acting elements of the chestnut *CmCER* gene family were made, mainly predicting hormone-response elements, stress-response elements and MYB-binding progenitors. The results of the prediction analysis showed an abundance of cis-acting elements in the chestnut *CmCER* gene family, which may determine the ability of *CmCER* genes to respond under stress conditions.

Chinese chestnut seedlings of different lengths treated under drought conditions were tested by qRT-PCR; 32 genes were activated, with the expression pattern of *CmCER* genes divided into three trends: upregulation, upregulation followed by downregulation, and then continuous downregulation. However, Qi et al. found that the expression of MdCER3, MdCER5, MdCER6 and MdCER9 showed downregulation followed by upregulation trends at 0 H, 3 H and 12 H in apple tissue culture seedlings treated with PEG. Drought treatment of three sunflower genotypes to study the expression pattern of HanCER10 and HanCER60 showed that plants with different genotypes have different drought tolerance. In passion fruit, the expression profiles of PeCERs were explored mainly in terms of different fruit developmental stages and different genes in the tissues. Similarly, Rizwan et al. found that the expression of PeCERs in the two passion fruit varieties was upregulated in the roots, stocks and leaves under drought stress. For the semi-thin section analysis of varieties in both the Beijing and Jiangsu germplasm repositories, results from the wild samples showed that the cuticle thickness of the northern samples was thicker than that of the southern samples, which may be due to long-term environmental selection resulting in different genotypes in the southern and northern samples. The expression results of wild Y-2 varieties in the south and Y-2 material grafted for one year in the north showed significant differences, with the more arid northern environment causing environmental induction of *CER* gene expression and promotion of wax biosynthesis, resulting in thicker cuticle thickness.

Developments in plant genomics have revealed the genetic basis of traits, combining genome-wide association studies (GWAS), genome–environment associations (GEA), genome-wide selection scans (GWSS) and other modern analytical approaches to infer the genetic basis of adaptation to abiotic stresses using environmental variables. Genome–environmental association analyses can use gradient forest models to detect adaptive signals in species. Genome-wide association studies use phenotypic and genomic data to identify the genetic causes of variation in phenotypes. The combination of histology and a variety of modern analytical methods allows for the prediction of chestnut adaptation to climate change, and it may even be applied to genetic breeding for chestnut tolerance to abiotic stresses.

## 4. Materials and Methods

### 4.1. Plant Materials

This study collected wild Chinese chestnut leaves from Gansu, Shaanxi, Sichuan, Hubei, Anhui, Zhejiang, Jiangxi and Hunan and varieties of Chinese chestnut leaves were collected from Beijing Chestnut Experiment Station and Liyang Longtan forestry nursery. The wild Y-2 variety from Hubei was grafted onto the variety in the Beijing germplasm repository, and the leaves were collected. After the leaves were collected, they were washed before being placed in a fixative solution for making semi-thin sections. All samples were frozen in liquid nitrogen and stored at −80 °C.

### 4.2. Thickness Statistics of Wax on the Surface of Leaves of Wild and Cultivated Chestnut Species

Preparation of semi-thin sections of Chinese chestnut leaves was as follows: Chinese chestnut leaves were cut to the correct size, placed in fixative and vacuumed for 40 min. They were washed with PBS buffer solution four times and soaked once in deionized water for 30 min each. The leaves were dehydrated with gradients of 10%, 30%, 50%, 70%, 90% and 100% alcohol for 30 min each at room temperature. SolutionA: 100% ethanol was infiltrated at volume ratios of 1:3, 1:1 and 3:1, for 12 h for each step, and then the leaves were placed in pure SolutionA overnight at 4 °C. The samples were then placed in a mold to create a plastic block and glued to the base. The slides were cleaned and put on a film dryer heated to a temperature of 57 °C. Then, 1–2 drops of water were added to the glass slides and they were sliced with a slicer (Reichert-Jung, Leica microsystems, Netherlands) to a thickness of 5 μm. The slices were placed on the slides, unfolded naturally and dried on the dryer. Finally, the sections were stained with Oil Red O dye [43] according to the instructions and photographed under a Leica (LeicaAu5500B) microscope. This was replicated for each variety three times; the epicuticular wax thickness was measured using ImageJ software, and analysis of significant differences was carried out using prism6 software. For quantitative data tests, one-way ANOVA was used, and Bonferroni’s multiple comparisons test was used for multiple comparisons between groups.

### 4.3. Sample Sequencing

The 22 samples sequenced in this study are listed in Appendix A. Genomic DNA was extracted from leaf samples using CTAB method. Libraries were constructed using 2.5 ug of high-quality DNA per sample. Sequencing libraries were generated using the Truseq Nano DNA HT Sample Prep Kit (Illumina, San Diego, CA, USA) and genome-wide paired-end reads were generated using the Illumina platform (HiSeq2000, Illumina, San Diego, CA, USA).

### 4.4. Selective Sweep

First, SNPs with minor allele frequency below 5% were removed from this analysis. To identify potential selective sweeps between northern samples and southern samples, FST was calculated using VCFtools with a 50 kb sliding window and a step size of 10 kb.

### 4.5. Identification of CER Gene Family Members and Analysis of Physicochemical Properties of Protein Sequences in Chinese Chestnut

The annotated *CER* sequences of *A. thaliana* were downloaded from the Arabidopsis genome annotation website (https://www.arabidopsis.org/, accessed on 11 June 2022), and the Chinese chestnut genome data were blasted with BioEdit software, and the E value threshold was 0.001 for candidate gene family. Then, the candidate genes domain sequences were obtained from CER protein sequence alignment analysis using NCBI-CDD database (https://www.ncbi.nlm.nih.gov/Structure/cdd/wrpsb.cgi, accessed on 11 June 2022).

On-line analysis software ExPASy (https://web.expasy.org/protparam/, accessed on 16 June 2022) was used for analysis of CER genes encoding amino acid number, isoelectric point (pI), molecular weight (MW), and other physical and chemical information. Then, the protein subcellular localization prediction software Cell-PLoc2.0 (http://www.csbio.sjtu.edu.cn/bioinf/Cell-PLoc-2/, accessed on 16 June 2022) was used for the *CmCER* family protein sequence analysis of subcellular localization prediction.

### 4.6. Phylogenetic Analysis

Multiple sequence alignment of *CER* amino acid sequences for *A. thaliana* and *C. mollissima* was performed using DNAMAN software, and the repeated sequences were removed. Then, MEGA11.0 software was used to determine the optimal conformational tree model of the *CER* gene family for *A. thaliana* and *C. mollissima*. Bootstrap = 1000 to construct the evolutionary tree. The iTOL online website (https://itol.embl.de/itol.cgi, accessed on 16 June 2022) was used to beautify the evolutionary tree.

### 4.7. Gene Structure and Motif Analysis

The Chinese chestnut *CER* gene protein sequences into NCBI-CDD database website (https://www.ncbi.nlm.nih.gov/Structure/cdd/wrpsb.cgi, accessed on 16 June 2022) to obtain the results file, and we then used the TBtools software for structure visualization with domain analysis.

The MEME (https://meme-suite.org/meme/tools/meme, accessed on 16 June 2022) website was used for Chinese chestnut *CER* genes encoding protein sequences with the conservative base sequence analysis. The gene structure and motifs of Chinese chestnut *CER* genes were visualized using TBtools software.

### 4.8. Chromosome Localization and Cis-Acting Element Analysis

The location of *CER* genes on the chromosomes of *A. thaliana* and *C. mollissima* was studied using the MapGene2Chromosome V2 website (http://mg2c.iask.in/mg2c_v2.1/, accessed on 22 June 2022)by gene ID, start position, stop position, chromosome length, etc.

To analyze the cis-acting elements contained in the promoter region, the upstream 2000 bp sequences of 34 Chinese chestnut *CER* genes were extracted using TBtools. The online software PlantCARE (http://bioinformatics.psb.ugent.be/webtools/plantcare/html/, accessed on 27 June 2022)was used to predict the cis function components.

### 4.9. qRT-PCR Assay

For the drought treatment, Chinese chestnut seedlings were removed from the pot, cleaned and gently wiped with paper towels for moisture, and placed on dry filter paper. Samples were collected at 0, 1, 3, 6, 9 and 12 h of treatment, and then flash-frozen in liquid nitrogen and stored at −80 °C for later use.

RNA was extracted from the samples using the RNA extraction kit Plant RNA Kit (Omega, Norcross, GA, USA). The elimination of genomic DNA was through treatment with RapidOut DNA Removal Kit (Thermo Scientific, Vilnius, Lithuania). The cDNA was synthesized using reverse transcriptase M-MLV (TaKaRa, Dalian, China). Primers were designed using Primer 5 software (Appendix A) and analyzed by real-time PCR using Super Real Pre MixPlus Kit (TaKaRa, Dalian, China) with SYBR Green. The reaction system was performed according to the instructions. The reaction instrument was CFX (Bio-Rad, Roche Diagnostics GmbH, Mannheim, Germany). The reaction system was as follows: template 2 μL, 10 μmol/L primer 0.5 μL, SYBR 5 μL, RNase free H_2_O 2.5 μL. Procedure: pre-denaturation at 95 °C for 2 min; denaturation at 95 °C for 15 s, annealing at 60 °C for 30 s, extension at 72 °C for 30 s, 40 cycles in total. *CmActin* was used as the reference gene, and for each sample this was performed in triplicate. The relative expression of *CmCERs* was calculated using the 2^−ΔΔCt^ method.

## 5. Conclusions

In this study, a significant difference in wax thickness in Chinese chestnut between northern and southern China was identified. Through genome-wide selective sweep analysis of two populations in the northwest and southeast, significant genetic differentiation was identified for *CmCER1-1*, *CmCER1-5* and *CmCER3*. We then identified 34 *CER* genes in the Chinese chestnut genome and analyzed their physicochemical properties, evolutionary relationships, gene structure and expression patterns. Drought treatment of Chinese chestnut seedlings affected the expression of *CmCER* genes. The expression of *CmCER1-1*, *CmCER1-5* and *CmCER3* differed significantly in the same Chinese chestnut variety cultivated in different locations. This provides a basis for functional studies on Chinese chestnut wax biosynthesis in relation to environmental adaptation.

## Figures and Tables

**Figure 1 ijms-23-16202-f001:**
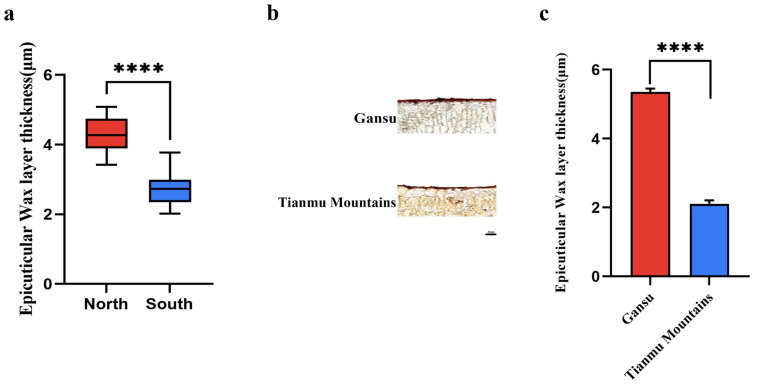
(**a**) Comparison of differences in leaf epicuticular wax layer thickness between northern and southern wild samples. Error bars indicate standard errors. ‘****’ on error bar indicates a significant difference between the two groups at *p* < 0.0001. (**b**) Characterization of the anatomical structures of Gansu and Tianmu Mountains samples. The length of the scale is 20μm. The complete semi-thin section is shown in Appendix A. (**c**) Leaf epicuticular wax layer thickness analysis of Gansu and Tianmu Mountains samples. Error bars indicate standard errors. ‘****’ on error bar indicates a significant difference between the two groups at *p* < 0.0001.

**Figure 2 ijms-23-16202-f002:**
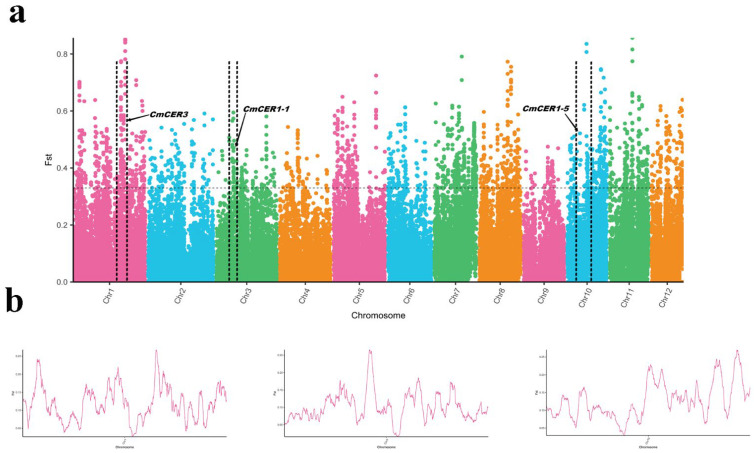
(**a**) Genomic landscape of the population differentiation in groups I (Gansu) and II (Tianmu Mountains) of wild Chinese chestnut, as measured by the fixation index (FST). (**b**) Sliding window analysis for further presentation.

**Figure 3 ijms-23-16202-f003:**
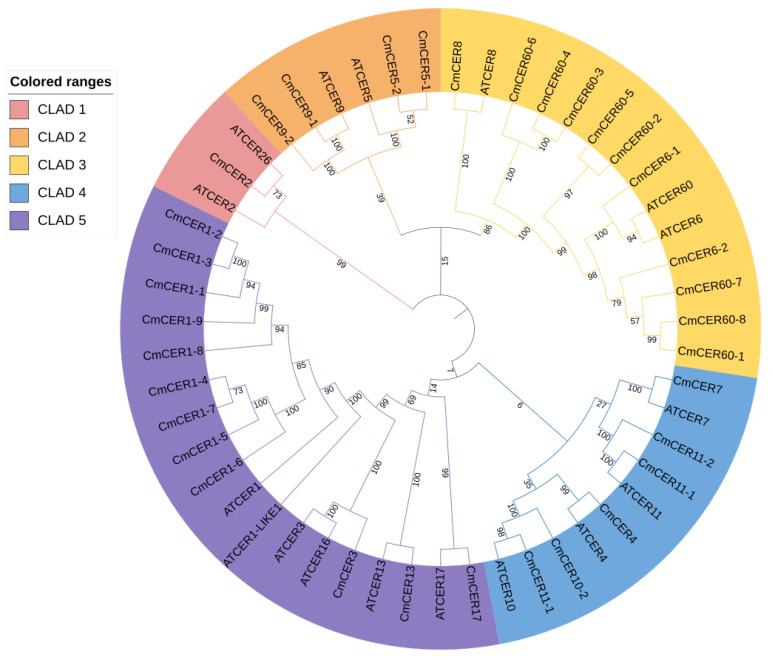
Phylogenetic trees of 17 *CERs* in *A. thaliana* and 34 CERs in *C. mollissima*. MEGA11 was applied to construct the maximum-likelihood (ML) tree using the WAG with Freqs. (+F) model with 1000 bootstrap replicates. The number on the branch indicates the bootstrap value. The different groups are color-coded as indicated in the top left panel.

**Figure 4 ijms-23-16202-f004:**
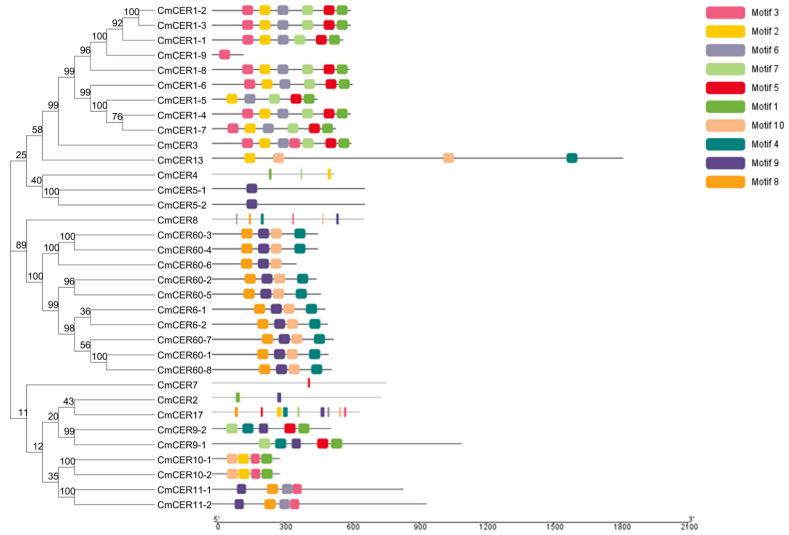
Distribution of conserved motifs in *CmCERs*. Individual motifs are indicated in different-colored horizontal bars below each figure. Numbers on the X-axis indicate amino-acid positions of the proteins.

**Figure 5 ijms-23-16202-f005:**
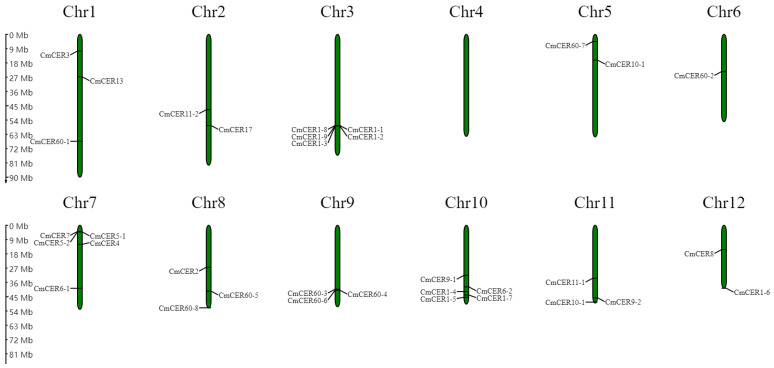
Chromosomal location of *CmCERs*.

**Figure 6 ijms-23-16202-f006:**
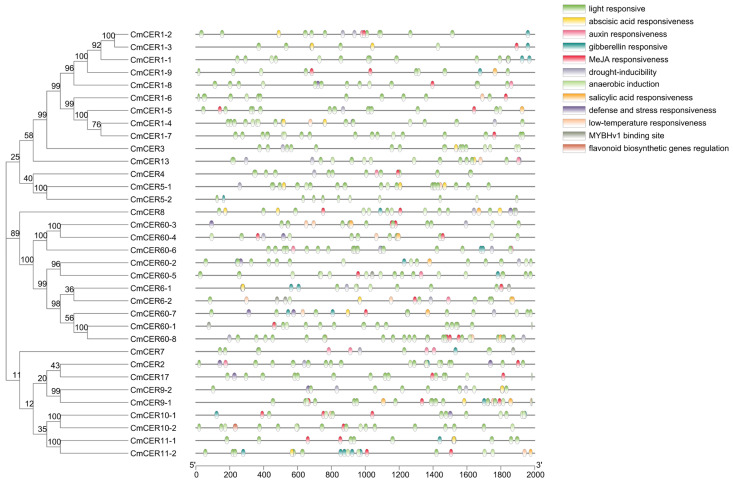
Cis-regulatory element analysis on *CmCER* genes.

**Figure 7 ijms-23-16202-f007:**
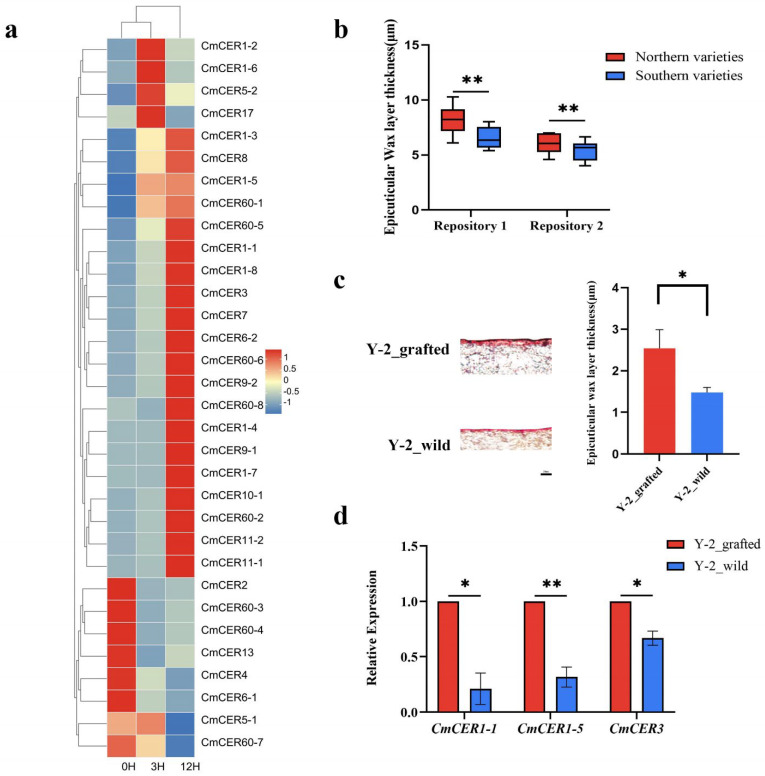
(**a**) Expression levels of *CmCER* genes in drought-treated Chinese chestnut seedlings. (**b**) Comparison of differences in leaf epicuticular wax layer thickness between northern and southern varieties in the Beijing and Liyang germplasm repositories. Repository 1: Chestnut Experiment Station in the Huairou District of Beijing, China. Repository 2: Liyang Longtan Linchang. Error bars indicate standard errors. ‘**’ on the error bar indicates a significant difference between the two groups at *p* < 0.01. (**c**) Results of semi-thin sections of wild Y-2 leaves in Hubei and grafted Y-2 leaves in the north. Error bars indicate standard errors. ‘*’ on the error bar indicates a significant difference between the two groups at *p* < 0.05. The length of the scale is 20μm. The complete semi-thin section is shown in Appendix A. (**d**) Expression of *CmCER1*-*1*, *CmCER1-5* and *CmCER3* in grafted Y-2 varieties in the north and wild Y-2 varieties in the south. Error bars indicate standard errors. ‘*’ or ‘**’ on the error bar indicates a significant difference between the two groups at *p* < 0.05 or *p* < 0.01, respectively.

**Table 1 ijms-23-16202-t001:** Physicochemical properties of *CER* genes identified in *C. mollissima*.

Gene Name	Gene ID	Number of Amino Acids	Molecular Weight/kD	Theoretical pI	Instability Index (II)	Aliphatic Index	Subcellular Localization
*CmCER1-1*	Cm03G01983	588	68.88	8.97	33.52	94.18	Vacuole
*CmCER1-2*	Cm03G01987	623	72.82	8.58	34.99	96.42	Vacuole
*CmCER1-3*	Cm03G01992	623	72.77	8.58	34.42	96.26	Vacuole
*CmCER1-4*	Cm10G01890	623	71.85	8.72	34.72	99.49	Vacuole
*CmCER1-5*	Cm10G02076	474	54.37	9.02	35.86	96.22	Cell membrane Vacuole
*CmCER1-6*	Cm12G02161	632	73.08	8.03	31.51	97.74	Vacuole
*CmCER1-7*	Cm10G01980	557	63.69	8.85	33.95	98.13	Vacuole
*CmCER1-8*	Cm03G01976	619	72.19	9.18	34.23	93.51	Vacuole
*CmCER1-9*	Cm03G01984	142	16.50	6.68	42.9	97.61	Vacuole
*CmCER2*	Cm08G01333	446	49.80	5.28	32.03	88.25	Cytoplasm
*CmCER3*	Cm01G00496	628	71.94	8.47	31.96	102.72	Vacuole
*CmCER4*	Cm07G00737	490	55.19	8.36	25.77	104.22	Golgi apparatus
*CmCER5-1*	Cm07G00263	696	77.79	9.35	48.11	93.99	Cell membrane
*CmCER5-2*	Cm07G00264	682	75.88	8.67	43.98	99.5	Cell membrane
*CmCER6-1*	Cm07G01950	496	55.85	9.01	40.14	99.11	Cytoplasm
*CmCER6-2*	Cm10G01697	507	57.07	9.29	37.49	94.97	Cytoplasm
*CmCER7*	Cm07G00259	457	49.88	5.94	53.18	76.21	Chloroplast Mitochondrion
*CmCER8*	Cm12G00788	654	73.85	6.36	29.42	86.73	Peroxisome
*CmCER9-1*	Cm10G01351	1109	123.64	5.85	36.29	109.1	Cell membrane Nucleus
*CmCER9-2*	Cm11G02224	534	58.97	6.11	35.25	106.84	Cell membrane
*CmCER10-1*	Cm05G00914	308	36.04	9.61	41.52	86.66	Nucleus
*CmCER10-2*	Cm11G02385	310	36.31	9.61	48.66	90.23	Cell membrane Golgi apparatus
*CmCER11-1*	Cm11G01507	854	95.40	6.35	51.84	80.68	Nucleus
*CmCER11-2*	Cm02G02052	952	106.28	5.93	56.97	80.09	Chloroplast Nucleus
*CmCER13*	Cm01G01426	1852	206.05	5.84	46.55	111.02	Nucleus
*CmCER17*	Cm02G02326	389	44.47	9.64	46.71	86.97	Chloroplast Endoplasmic reticulum
*CmCER60-1*	Cm01G03190	510	57.62	9.08	42.32	94.63	Chloroplast Cytoplasm
*CmCER60-2*	Cm06G01231	457	50.95	8.74	37.59	97.72	Cytoplasm
*CmCER60-3*	Cm09G01848	463	52.35	8.69	43.47	93.89	Cytoplasm
*CmCER60-4*	Cm09G01872	463	52.35	8.69	43.47	93.89	Cytoplasm
*CmCER60-5*	Cm08G02023	478	53.64	9.71	34.43	99.96	Cytoplasm
*CmCER60-6*	Cm09G01873	370	42.02	9.58	40.14	98.54	Chloroplast Cytoplasm
*CmCER60-7*	Cm05G00220	532	60.17	8.99	32.68	90.55	Cytoplasm
*CmCER60-8*	Cm08G02594	523	59.02	9.27	39.35	92.64	Cytoplasm

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
