# Peer review of "Genome-Wide Identification of the CER Gene Family and Significant Features in Climate Adaptation of Castanea mollissima"

_ijms, 2022, doi:10.3390/ijms232416202_

Round 1

Reviewer 1 Report

To,

The Editor,

IJMS, MDPI,

Manuscript ID: ijms-2033615

 Subject: Submission of comments of the manuscript in “IJMS"

 Dear Editor IJMS, MDPI,

 Thank you very much for the invitation to consider a potential reviewer for the manuscript (ID: ijms-2033615). My comments responses are furnished below as per each reviewer’s comments. 

In the reviewed manuscript, authors identified 44 CER genes in the chestnut genome. Further, identified 34 CmCER genes were analyzed for their physicochemical properties, phylogenetic tree construction, conserved motif analysis, chromosome positioning and cis-element prediction. Drought stress treatment of chestnut seedlings resulted in three expression patterns of CmCER genes, with most CmCER gene expression levels being up-regulated, five CmCER genes showing an upward and then downward trend, and seven CmCER genes showing a continuous downward trend in expression levels. Analysis of varieties from two resource repositories in Beijing and Jiangsu showed that the cuticle thickness of the northern variety was thicker than that of the southern variety. For the Y-2, expression analyses of northern and southern, grafted annual material on the same rootstock showed significant differences in the expression of CmCER1-1, CmCER1-5 and CmCER3. These findings lay the foundation for further exploration of the CmCER genes and the importance of studying the adaptation of chestnut wax biosynthesis to the southern and northern environments. In general, the manuscript represents a very big piece of information. Therefore, it might be conditionally accepted subject to major revision. Authors have to improve their manuscripts with many non-clear meanings, inaccuracies and inconsistencies, and the authors need to address the following issues before it can be accepted for publication. 

  1. I have read the entire manuscript and my initial comment is that manuscript is poorly written. I have significant concerns about the grammar and vocabulary of the manuscript; therefore, improvement of the language is highly needed. 
  2. The structure of the abstract should be improved, as well as the lack of several aspects that should be included in this section. Most of the abstracts contain confusing and uninformative sentences. Please give more precise objectives here (such as in the Abstract). 
  3. Line no-26, what does mean by Y-2?
  4. Introduction grammatical issues appear to be most prevalent in the introduction, making for very confusing reading. Further, the introduction is short but has no clear thread. 
  5. General note: the figures in this section are quite low resolution and difficult to make out. Higher-resolution versions will be needed for publication, for example, in Figures 2, 5, 6, and Figure S2.
  6. All Figure legends must be self-explanatory.
  7. Discussion- many times references are made to the information given in the Introduction section (sometimes more general information). It would be good to discuss especially the results and critically, ie. Which can cause differences in the results of authors and other articles.
  8. I would like the Authors to provide the methodology and results of the number of replications wherever possible. The same applies to the statistical significance of the results. Please describe statistical methods used in the work in materials and methods.
  9. The qRT-PCR methodology provided is also very vague and confusing. Please provide more details like the calibrator used in the study. I assume the authors have used the control as the calibrator. If so, the authors should not include the control within the bar graph as it represents the fold change between the treated vs control and a fold change of “1” for the ‘control’ doesn’t make any sense.  Also, would be good to provide details on what reagents (details of probes used, if any, if SYBR was used then details for that, etc.) and real-time PCR machines were used in the current study.
  10. The conclusion section is very poorly written. It should be extensively improved.
  11. References: shall have to correct the whole References according to the ”Instructions for the Authors”, e.g. title should not be in italics, the Journal name is in italics, and the author shall have to use the abbreviated name Journals cited the year must be bold, the scientific name must be italics etc. Please check all references carefully.
  12. L-475 Hlianthus Annuus was replaced with Hlianthus Annuus.
  13. L-478 Ziziphus Jujube replaced with Ziziphus Jujube etc

Author Response

Dear reviewers and editors:

Many thanks to the reviewers and editors for their comments on this paper. We have carefully considered the suggestion of reviewer and make some changes. According to the questions, we have made corrections and explanations and highlighted the text of the revised part (red font) for reference. Responds to the reviewers' comments:

Question 1:

I have read the entire manuscript and my initial comment is that manuscript is poorly written. I have significant concerns about the grammar and vocabulary of the manuscript; therefore, improvement of the language is highly needed.

Response:

Thanks to the reviewers' comments, the MDPI team (https://www.mdpi.com/authors/english.) has been chosen for the clearer English editing of this manuscript in response to language and vocabulary issues.

Question 2:

The structure of the abstract should be improved, as well as the lack of several aspects that should be included in this section. Most of the abstracts contain confusing and uninformative sentences. Please give more precise objectives here (such as in the Abstract).

Response:

The suggestion was so valuable to the paper that we have reorganized the abstract in line with the reviewers' comments. The Abstract is written again as follows:

The plant cuticle is the outermost layer of the aerial organs and an important barrier against biotic and abiotic stresses. The climate varies greatly between the north and south of China, with large differences in temperature and humidity, but Chinese chestnut is found in both regions. This study investigated the relationship between the wax layer of chestnut leaves and environmental adaptation. Firstly, semi-thin sections were used to verify that there is a significant difference in the thickness of the epicuticular wax layer between wild chestnut leaves in northwest and southeast China. Secondly, a whole-genome selective sweep was used to resequence wild chestnut resources from two typical regional populations, and significant genetic divergence was identified between the two populations in the CmCER1-1, CmCER1-5 and CmCER3 genes. Thirty-four CER genes were identified in the whole chestnut genome, and a series of predictive analyses were performed on the identified CmCER genes. The expression patterns of CmCER genes were classified into three trends—upregulation, upregulation followed by downregulation, and continuous downregulation—when chestnut seedlings were treated with drought stress. Analysis of cultivars from two resource beds in Beijing and Liyang showed that the wax layer of the northern variety was thicker than that of the southern variety. For the Y-2 (Castanea mollissima genome sequencing material) cultivar, there were significant differences in the expression of CmCER1-1, CmCER1-5 and CmCER3 between the southern variety and the northern one-year grafted variety. Therefore, this study suggests that the CER family genes play a role in environmental adaptations in chestnut, laying the foundation for the further exploration of CmCER genes. It also demonstrates the importance of studying the adaptation of Chinese chestnut wax biosynthesis to the southern and northern environments. See Lines 135-148 for details

Question 3:

Line no-26, what does mean by Y-2?

Response:

Y-2 is the plant material that we did genome sequence of Castanea mollissima (Xing et al. 2019).  

Question 4:

Introduction grammatical issues appear to be most prevalent in the introduction, making for very confusing reading. Further, the introduction is short but has no clear thread.

Response:

Thanks for the reviewer's attention to this content. In the case of the English editing of the manuscript, we have rewritten the introduction and restating the idea of the article at the end of the introduction. Please see the introduction part and the Lines135-148 for details.

Question 5:

General note: the figures in this section are quite low resolution and difficult to make out. Higher-resolution versions will be needed for publication, for example, in Figures 2, 5, 6, and Figure S2.

Response:

Figures 2, 5, 6, and Figure S2 are replaced with higher-resolution images as the journal’s required

Question 6:

All Figure legends must be self-explanatory.

Response:

We have modified the legend to make it more explicit.

Question 7:

Discussion- many times references are made to the information given in the Introduction section (sometimes more general information). It would be good to discuss especially the results and critically, ie. Which can cause differences in the results of authors and other articles.

Response:

The suggestion was so valuable to the paper that we have reorganized the discussion in line with the reviewers' comments. As the reviewers’ suggestion, the Discussion is rewritten and some comparison with other results of authors and other articles was also added in the revision of MS. Please see L356-370, L411-419 for details.

Question 8:

I would like the Authors to provide the methodology and results of the number of replications wherever possible. The same applies to the statistical significance of the results. Please describe statistical methods used in the work in materials and methods.

Response:

We have added the statistical methods used in our work to Materials and Methods as follows: Each variety was replicated three times, the epicuticular wax thickness was measured using ImageJ software, and analysis of significant differences was carried out using prism6 software. For quantitative data tests, one-way ANOVA was used, and Bonferroni's multiple comparisons test was used for multiple comparisons between groups. See L449-453 for details.

Question 9:

The qRT-PCR methodology provided is also very vague and confusing. Please provide more details like the calibrator used in the study. I assume the authors have used the control as the calibrator. If so, the authors should not include the control within the bar graph as it represents the fold change between the treated vs control and a fold change of “1” for the ‘control’ doesn’t make any sense. Also, would be good to provide details on what reagents (details of probes used, if any, if SYBR was used then details for that, etc.) and real-time PCR machines were used in the current study.

Response:

Thanks for the reviewer's comments. We have re-written the qRT-PCR method in Materials and Methods, adding reagents, details of the real-time PCR machines, etc, as follows: RNA was extracted from the samples using the RNA extraction Kit Plant RNA Kit (Omega, Norcross, GA, USA). The elimination of genomic DNA through treatment with RapidOut DNA Removal Kit (Thermo Scientific, Vilnius, Lithuania). The cDNA was synthesized using Reverse transcriptase M-MLV (TaKaRa). Primers were designed using Primer 5 software (Table S2) and analyzed by real-time PCR using Super Real Pre MixPlus Kit (TaKaRa) with SYBR Green. The reaction system was performed according to the instructions. The reaction instrument was CFX (Bio-Rad). The reaction system is as follows: template 2μl, 10μmol /L primer 0.5μl, SYBR 5μl, RNase free H2O 2.5 μl. Procedure: pre-denaturation at 95℃ for 2 min; denaturation at 95℃ for 15s, annealing at 60℃ for 30s, extension at 72℃ for 30 s, 40 cycles in total. CmActin was used as the reference gene, and each sample was performed in triplicate. The relative expression of CmCERs was calculated using the 2-ΔΔCt method. See L507-518 for details.

Question 10:

The conclusion section is very poorly written. It should be extensively improved.

Response:

Thanks for the reviewer's comments. We have made improvements to the conclusion section, as follows: In this study, a significant difference in wax thickness in Chinese chestnut be-tween the northern and southern China was identified through the material collected by the group from the north and south wild and resource beds. Through genome-wide selective sweep analysis for of two populations in the northwest and southeast, significant genetic differentiation was identified for CmCER1-1, CmCER1-5 and CmCER3. We then identified 34 CER genes in the Chinese chestnut genome and analyzed their phys-icochemical properties, evolutionary relationships, gene structure and expression pat-terns. Drought treatment of Chinese chestnut seedlings affected the expression of CmCER genes. The expression of CmCER1-1, CmCER1-5 and CmCER3 differed significantly in the same Chinese chestnut variety cultivated in different locations. This provides a basis for functional studies on Chinese chestnut wax biosynthesis in relation to environmental adaptation

Question 11:

References: shall have to correct the whole References according to the “Instructions for the Authors”, e.g. title should not be in italics, the Journal name is in italics, and the author shall have to use the abbreviated name Journals cited the year must be bold, the scientific name must be italics etc. Please check all references carefully.

Response:

Thank you very much for the conscientiousness of the reviewers. The format of the references has been carefully checked and revised.

Question 12-13:

L-475 Hlianthus Annuus was replaced with Hlianthus Annuus.

L-478 Ziziphus Jujube replaced with Ziziphus Jujube etc

Response:

Changes were made to questions 12-13, and similar questions were checked and amended.

Reviewer 2 Report

The work by Zhao et al. explored the genomic composition of 34 CER genes is Chinease chestnut using comparative bioinformatics. As a second step, authors induced these CER genes by drought stress, offering insight into natural adaptation of chestnut trees to the northern and southern climates of China. This is a pertinent work, which addresses the genetic determinants of a key trait to improve abiotic stress tolerance in tree species with a bio-economical potential for Asia. The work is well written, statistically up to date, and highlights key findings in the genetic architecture of adaptation is chestnut trees.

First, I recognize authors for mentioning in L92 the broad consequences of abiotic stresses in tree species, including drought. Still, I am missing key references regarding the polygenetic nature of the genomic architecture for abiotic stresses and drought that may be pleiotropic with more complex effects beyond the CER gene family. Specifically, other gene families have also been linked with correlated abiotic stresses and drought tolerance, as indicated in: (i) Plant Science 2016 242:250 for the ERECTA gene family in association with the AP2 domain, (ii) BMC Genetics 2012 13:58 for the ASR family f in association with the ABA-dependent MYB, and (iii) Theor Appl Genet 2012 125(5):1069-85 for the DREB transcription factor family pleotropic in several pathways with the WRKY transcription factor. Authors should discussed these cases explicitly and make a clear point since the introduction on why focusing only on the CER gene family. Please revisit this point at the discussion and recommend expanding the analysis to other gene families.

Second, my major analytical suggestion is to replace the per-marker Fst profile in figure 2 (L152) by an averaged Fst across sliding windows (refer to figure 3 in Front Plant Sci 2018 9:1816, to be included). Sliding windows offer a less noisy approach for the inference of selective footprints across the genomes. Also, couple the sliding window approach with other statistics beyond Fst, such as delta divergence and Dxy (refer to figure 4 in Front Plant Sci 2018 9:1816, to be included).

Thir, please also complement the genomic landscape inference in L152 (figure 2) with explicit gene-environment associations, which have higher power for the detection of environmental effects compared to genomic scans of divergence (refer to the seminal review Front Genet 2022 13:910386). The gene-environment approach has been validated for heat (refer to Front Genet 2019 10:954) and drought (refer to Front Plant Sci 2018 9:128 and Genes 2021 12:556), stresses in which the authors are interested to interpret Castanea divergence.

Additionally, include in main figure 3 (L195) explicit numeric bootstrap values within the gene tree for support, as in figures 4 and 6 (symbols in figure 3 are very hard to follow). Also, authors should explicitly comments on the significance of the replication level utilized to gather the expression profiles, for which a preliminary power analysis would be insightful.

As closure, please include a perspective section at the end of the introduction in L347 with recommendation on how to better integrate omic technologies with modern analytical approaches to assess adaptation and abiotic stress tolerance in tree species by referring to the seminal reviews Front Plant Sci 2020 11:583323 and Front Genet 2020 11:564515, and its link with other type of stresses in forest trees (e.g. biotic pressures as discussed in Plants 2021 10:2022). For sure future studies on adaptation in chestnut trees would benefit from these fresh innovative perspectives.

Finally, in terms of writing, the abstract is very poor and does not follow the ABT recommendation (see this card: https://entomologychallenges.files.wordpress.com/2018/10/abt-shorthand-reference-card.pdf) for abstracts (in L12). Please implement them.

Related with the previous point, please enlist explicit research gap, research hypotheses and goals at the end of the introduction (L121). This will allow readers focusing on explicit expectation when approaching the report.

Author Response

Dear reviewers and editors:

Many thanks to the reviewers and editors for their comments on this paper. We have carefully considered the suggestion of reviewer and make some changes. According to the questions, we have made corrections and explanations and highlighted the text of the revised part (red font) for reference. Responds to the reviewers' comments:

Question 1:

First, I recognize authors for mentioning in L92 the broad consequences of abiotic stresses in tree species, including drought. Still, I am missing key references regarding the polygenetic nature of the genomic architecture for abiotic stresses and drought that may be pleiotropic with more complex effects beyond the CER gene family. Specifically, other gene families have also been linked with correlated abiotic stresses and drought tolerance, as indicated in: (i) Plant Science 2016 242:250 for the ERECTA gene family in association with the AP2 domain, (ii) BMC Genetics 2012 13:58 for the ASR family f in association with the ABA-dependent MYB, and (iii) Theor Appl Genet 2012 125(5):1069-85 for the DREB transcription factor family pleotropic in several pathways with the WRKY transcription factor. Authors should discussed these cases explicitly and make a clear point since the introduction on why focusing only on the CER gene family. Please revisit this point at the discussion and recommend expanding the analysis to other gene families.

Response:

Thanks for the comments from the reviewers. In response to the suggestions given by the reviewers, we have carefully read the literature given by the reviewers and added the following to the discussion section: Drought tolerance is the ability of a plant to tolerate low water potential and maintain a certain level of physiological activity, growth and development. It is a complex quantitative trait in which the plant interacts with the drought environment. Drought tolerance in plants is influenced by several gene families; the ERECTA proteins belong to the protein kinase superfamily and the serine/threonine protein kinase family, which influence plant drought tolerance by affecting stomatal development, pathogen defense and phytohormone perception [47] . The DREB genes, which belong to the AP2/ EREBP family, are involved in the control of non-ABA-dependent drought stress responses in combination with plant-specific, stress-regulated transcription factors [48] . The Asr gene family plays an important role in plant adaptation to drought, especially the Asr1 gene, which is expressed in a housekeeping manner, and small changes in the protein can have a serious impact on plant adaptation[49] . Although other gene families have also been associated with drought tolerance, in this study, the focus on the CER gene family was due to the significant divergence of CERs genes between the NW and SE populations identified in the whole genome resequencing sweep analysis. See Lines 352-366 for details.

Question 2:

Second, my major analytical suggestion is to replace the per-marker Fst profile in figure 2 (L152) by an averaged Fst across sliding windows (refer to figure 3 in Front Plant Sci 2018 9:1816, to be included). Sliding windows offer a less noisy approach for the inference of selective footprints across the genomes. Also, couple the sliding window approach with other statistics beyond Fst, such as delta divergence and Dxy (refer to figure 4 in Front Plant Sci 2018 9:1816, to be included).

Response:

The suggestion was so valuable to the paper. We have revised Figure 2 as suggested by the reviewer.

.Question 3:

Thir, please also complement the genomic landscape inference in L152 (figure 2) with explicit gene-environment associations, which have higher power for the detection of environmental effects compared to genomic scans of divergence (refer to the seminal review Front Genet 2022 13:910386). The gene-environment approach has been validated for heat (refer to Front Genet 2019 10:954) and drought (refer to Front Plant Sci 2018 9:128 and Genes 2021 12:556), stresses in which the authors are interested to interpret Castanea divergence.

Response:

Thanks for the comments from the reviewers, this suggestion is also our concern in the analysis strategy. Here,There are some reasons: 1. The samples, we selected for analysis come from two distinct ecotypes of populations. The two populations have obvious local environmental adaptability through long-term natural selection. 2. In view of the spatial difference between the two populations, selective sweep analysis using Index of genetic differentiation between populations (Fst) may be a preferred method. Simultaneously, the total number of the two populations sequenced is 20. If environmental factor association analysis is used, its detection efficiency is very limited. (Generally, there should be more than 200 individuals for effective association analysis). So we use selective sweep analysis to identify these genes for local adaptation.

Question 4:

Additionally, include in main figure 3 (L195) explicit numeric bootstrap values within the gene tree for support, as in figures 4 and 6 (symbols in figure 3 are very hard to follow). Also, authors should explicitly comments on the significance of the replication level utilized to gather the expression profiles, for which a preliminary power analysis would be insightful.

Response:

We have modified Figure 3 by adding the numeric bootstrap values within the gene tree.

Question 5:

As closure, please include a perspective section at the end of the introduction in L347 with recommendation on how to better integrate omic technologies with modern analytical approaches to assess adaptation and abiotic stress tolerance in tree species by referring to the seminal reviews Front Plant Sci 2020 11:583323 and Front Genet 2020 11:564515, and its link with other type of stresses in forest trees (e.g. biotic pressures as discussed in Plants 2021 10:2022). For sure future studies on adaptation in chestnut trees would benefit from these fresh innovative perspectives.

Response:

Thank you very much for the conscientiousness of the reviewers. We have added an analysis of this section to the article, which reads as follows: Developments in plant genomics have revealed the genetic basis of traits, combining genome-wide association studies (GWAS) , genome-environment associations (GEA) , genome-wide selection scans (GWSS) and other modern analytical approaches to infer the genetic basis of adaptation to abiotic stresses using environmental variables. Genome-environmental association analyses can use gradient forest models to detect adaptive signals in species. Genome-wide association studies use phenotypic and genomic data to identify the genetic causes of variation in phenotypes. The combination of histology and a variety of modern analytical methods allows for the prediction of chestnut adaptation to climate change, and may even be applied to genetic breeding for chestnut tolerance to abiotic stresses. See Lines 416-425 for details.

Question 6:

Finally, in terms of writing, the abstract is very poor and does not follow the ABT recommendation (see this card: https://entomologychallenges.files.wordpress.com/2018/10/abt-shorthand-reference-card.pdf) for abstracts (in L12). Please implement them

Response:

The suggestion was so valuable to the paper that we have reorganized the abstract in line with the reviewers' comments. The Abstract is written again as follows: the plant cuticle is the outermost layer of the aerial organs and an important barrier against biotic and abiotic stresses. The climate varies greatly between the north and south of China, with large differences in temperature and humidity, but Chinese chestnut is found in both regions. This study investigated the relationship between the wax layer of chestnut leaves and environmental adaptation. Firstly, semi-thin sections were used to verify that there is a significant difference in the thickness of the epicuticular wax layer between wild chestnut leaves in northwest and southeast China. Secondly, a whole-genome selective sweep was used to resequence wild chestnut resources from two typical regional populations, and significant genetic divergence was identified between the two populations in the CmCER1-1, CmCER1-5 and CmCER3 genes. Thirty-four CER genes were identified in the whole chestnut genome, and a series of predictive analyses were performed on the identified CmCER genes. The expression patterns of CmCER genes were classified into three trends—upregulation, upregulation followed by downregulation, and continuous downregulation—when chestnut seedlings were treated with drought stress. Analysis of cultivars from two resource beds in Beijing and Liyang showed that the wax layer of the northern variety was thicker than that of the southern variety. For the Y-2 (Castanea mollissima genome sequencing material) cultivar, there were significant differences in the expression of CmCER1-1, CmCER1-5 and CmCER3 between the southern variety and the northern one-year grafted variety. Therefore, this study suggests that the CER family genes play a role in environmental adaptations in chestnut, laying the foundation for the further exploration of CmCER genes. It also demonstrates the importance of studying the adaptation of Chinese chestnut wax biosynthesis to the southern and northern environments.

Question 7:

Related with the previous point, please enlist explicit research gap, research hypotheses and goals at the end of the introduction (L121). This will allow readers focusing on explicit expectation when approaching the report.

Response:

Thanks for the reviewer's attention to this content. In the case of the English editing of the manuscript, we focused on editing the introduction and restating the idea of the article at the end of the introduction. A significant relationship between drought and CER genes has been confirmed in species such as tomato and grape, where CER genes influence wax biosynthesis, and in species such as apple, passion fruit and sunflower, for which the CER family was analyzed; however, the relationship between chestnut CER genes and drought stress has been less studied, and there is no systematic analysis of CER genes in Chinese chestnut. In the investigation of wild chestnut resources, we found that the thicknesses of the cuticles of the leaves in the north and south were different. At the same time, genes related to cuticle synthesis were found through the genome-wide selective sweep analysis between the northern and southern populations of wild chestnut. Therefore, in this study, 34 CER genes from the C. mollissima genome were systematically characterized and analyzed through comparative bioinformatics. It was found that CER gene expression was induced by drought stress. This study provides a basis for further research on wax biosynthesis in chestnut leaves. We explored the importance of the C. mollissima CER gene family in adaptation to northern and southern climates. see Lines135-148 for details.

Round 2

Reviewer 1 Report

Dear Editor,

Thank you for providing the opportunity to review the revised manuscript. The manuscript is improved considerably after revision according to the reviewer's comment. Now this study is a suitable contribution to the IJMS. I recommend the manuscript for publication.

Thank you

With best regards